# Challenges in detecting wind turbine power loss: the effects of blade erosion, turbulence and time averaging

Tahir H. Malik[1] and Christian Bak[2]

[1]Vattenfall, Amerigo-Vespucci-Platz 2, 20457, Hamburg, Germany
[2]DTU Wind and Energy Systems, Frederiksborgvej 399, 4000 Roskilde, Denmark
**Correspondence:** Tahir H. Malik (tahir.malik@vattenfall.de)

**Abstract.** Establishing a clear correlation between blade leading edge erosion (LEE) and the performance of operational wind turbines is challenging due to the complex interplay of various factors. This study aims to improve the understanding and analysis of real wind turbine measurements by employing aeroelastic simulations to investigate the combined effects of LEE, turbulence intensity ($TI$) and time averaging as a data processing technique and how they obscure the effects of erosion. Importantly, the study does not aim to investigate each contributing factor in detail but seeks to provide insights through selected examples, thereby illustrating how these conditions hinder the detection of blade erosion's effects on power loss. An offshore original equipment manufacturer (OEM) provided aeroelastic model was used to simulate various scenarios. Turbulence intensity was varied for a range of wind speeds and the aerofoil characteristics for the blade were modified to simulate different degrees of erosion, represented by varying levels of roughness. For a given site, findings reveal that even mild simulated erosion can reduce the annual energy production (AEP) by 0.82% at 6% $TI$, while more severe erosion leads to a 1.46% decrease. Furthermore, increasing $TI$ exacerbates these losses, with a 15% $TI$ causing up to a 2.14% AEP reduction for eroded blades, making it increasingly difficult to distinguish between the effects of blade erosion and turbulence intensity ($TI$) on turbine performance. These effects are most pronounced at sites with lower average wind speeds. Moreover, the interaction between $TI$ levels and longer time-averaging periods, which vary with wind speed, can obscure the true magnitude of LEE's impact on short-term power fluctuations. This study demonstrates that 10-minute time-averaging periods can significantly mask performance and that analysing unsteady rotor data with shorter time periods, such as 1-second periods, is preferable. The work emphasises the importance of considering blade condition's impact in the context of various influencing factors for accurate AEP assessments, performance monitoring and improved wind turbine design for operational wind turbines.

## 1 Introduction

The performance of wind turbines is a multifaceted subject of research, being intricately affected by a multitude of environmental (Wharton and Lundquist (2012)) and operational factors. Wind turbine manufacturers and owners place significant emphasis on this aspect due to its implications for revenue as well as operations and maintenance (O&M). Despite this, accurately identifying and validating performance within operational wind turbines using their self-generated supervisory control and data acquisition (SCADA) data remains a major challenge (Ding et al. (2022)). This challenge stems from the complex

interplay of factors affecting the turbine's performance (Barthelmie and Jensen (2010)), making it difficult to isolate the effects of individual causes amidst the numerous variables and uncertainties. Consequently, extensive efforts are invested in analysing SCADA data, with the default approach involving the analysis of 10-minute time averaged values of wind speed and power, focusing particularly on power degradation over time. Nevertheless, it is acknowledged that significant uncertainties exist within this 10-minute time averaging analysis (Yang et al. (2014)), complicating the detection of leading edge erosion's (LEE) effects.

In industrial practice, operators typically calculate power curve loss contributions using static components, employing static tables that include factors such as the thrust coefficient, $C_t$, temperature, wind shear, transformer losses and component friction. Yet, quantifying the impact of LEE on the power curve for operating turbines remains a challenge. Despite the extensive research on individual factors such as turbulence and other environmental conditions, a comparative analysis of blade erosion's impact relative to effects such as turbulence intensity and time averaging periods has remained unexplored for operational turbines, which the present study aims to address.

This study specifically investigates the degradation of power due to LEE. The detrimental effects of LEE or leading edge roughness (LER) on aerofoil characteristics have been extensively documented in wind tunnel experiments (Hansen (2008); Maniaci et al. (2016); Gaudern (2014); Krog Kruse et al. (2021); Bak et al. (2023)). Furthermore, these effects have also been the subject of numerous studies on the impact of erosion on wind turbine annual energy production (AEP) (Bak et al. (2016); Ehrmann et al. (2017); Kruse (2019); Han et al. (2018); Castorrini et al. (2023)). These studies indicate potentially significant AEP losses of up to 7%. While the impact of blade erosion on AEP is generally smaller than that of wake deficits and some controllers can compensate for degraded lift through pitch adjustments, its subtle effects are nonetheless crucial to quantify. This study employs multibody simulations to capture the interaction between LEE and factors including $TI$ and data time averaging, providing a more quantitative understanding of how these factors obscure performance losses in SCADA data, aiming to bridge the gap in understanding. Where, currently, a 1% variance in AEP for Vattenfall, an energy utility, equates to an average daily loss of approximately 380 MWh. Although the effects of LEE on aerodynamic performance are easily measurable in controlled environments such as wind tunnels, the question is not whether aerodynamic losses occur; instead, it is why these effects are obscured within the scattered sensor signals generated by operational wind turbines and how to detect them when a rotor operates in a turbulent flow field with significant wind fluctuations.

Analysis of extensive measurement data from wind farms revealed difficulties in obtaining sufficient insight into the influencing mechanisms, a finding supported by studies from Badihi et al. (2022) and Gonzalez et al. (2019). Consequently, simulations of a wind turbine within a wind farm environment were considered more valuable than solely studying SCADA data. The analysis of the simulated data, again, revealed that understanding how turbulence intensity ($TI$) and the effect of averaging unsteady data influenced the results was crucial for interpreting both measured and simulated data. Furthermore, turbulence is a well-known atmospheric condition that significantly impacts wind turbine performance (St. Martin et al. (2016); Saint-Drenan et al. (2020); Kim et al. (2021); Cappugi et al. (2021)).

This study aims to investigate selected factors that obscure the detection of erosion-induced power losses in operational wind turbines, an area that has not been extensively examined in previous research. Rather than conducting an exhaustive analysis of all potential contributors, the investigation focuses on providing insights into these obscuring effects through key examples and

proposes potential mitigation strategies. While the need for further analysis is acknowledged, the objective is to demonstrate how specific atmospheric conditions and analysis methods complicate the identification of blade erosion's impact on power loss. A distinctive aspect of this work is the incorporation of a certified model of an operational turbine's controller into a full aero-servo-elastic simulation loop, which ensures that the response to degraded blades, including pitch adjustments utilising aerodynamic reserves, is captured accurately.

In this manner the study aims to improve the understanding and analysis of wind turbine performance measurements, rather than focusing on aerodynamic computations. The goal is to develop more reliable methods for detecting degradation in real-world wind turbine performance. With these aims the study also investigates and compares significant effects, such as turbulence intensity, alongside the impact of degraded aerofoil polar coefficients ($C_l$ and $C_d$), to uncover why erosion's effects are not easily detected in SCADA data. The influence of turbulence intensity is investigated at the rotor level, expanding upon existing knowledge that primarily focuses on performance at the aerofoil level (e.g., Bak et al. (2008) and Cappugi et al. (2021)). Furthermore, the effects of time-averaging, traditionally performed using 10-minute periods, are examined.

## 2  Method

This study aims to conduct a fundamental investigation into the impact of turbulence intensity on the aerodynamic performance of wind turbine rotors, focusing on the effects of leading-edge erosion. This was achieved using an aeroelastic code that incorporates structural dynamics. The effects of wind shear were also briefly investigated. Additionally, the study examined the potential impact of different time-averaging periods used in operational data analysis on the ability to detect and quantify the effects of leading-edge erosion.

### 2.1  Wind turbine and aeroelastic code

The investigation utilised the blade element momentum (BEM) based multi-body aero-servo-elastic tool HAWC2, developed by DTU Wind Denmark. A comprehensive description, usage and implementation of HAWC2 is well-documented in the literature Larsen and Hansen (2007). The certified multibody model used in this study, provided by an OEM, represents a currently operational offshore wind turbine. It is a three-bladed, multi-megawatt, horizontal axis wind turbine with variable speed, pitch regulation and yaw control with a nominal power between 3 and 4 MW. The Reynolds number, $Re$, can be estimated using the rule of thumb from Bak (2023), which states that $Re$ is proportional to the rotor radius, $R$ and ranges between $75,000 \cdot R$ and $150,000 \cdot R$. Consequently, $Re$ is approximately 7 million. Due to intellectual property considerations, specific details about the turbine, such as structural properties and control philosophy, are not disclosed; hence, the power is presented as *normalised power* and is expressed as power relative to the rated power.

While reference wind turbines such as the NREL 5 MW (Jonkman et al. (2009)) or the DTU 10 MW (Bak et al. (2013)) could have been employed, this study's close connection to wind farm measurements necessitated incorporating a controller from an actual wind turbine to investigate unsteady effects. Since relative changes in performance are more critical than absolute performance, analysing a real wind turbine model was considered essential. Various parameters, such as damage severity, radial

position and the turbine-specific power, impact potential degradation. Therefore, this study is expected to indicate general trends, with specific numerical results likely to vary slightly depending on the actual wind turbine design.

## 2.2 Representing leading edge erosion

Blade leading edge erosion was modelled as varying levels of surface roughness, a quantifiable measure of damage severity directly impacting aerodynamic performance and representing a precursor to more significant aerofoil deterioration where voids or cavities may begin to form. The multibody model's original blade aerofoil polars for the outer 15% of blade length were modified applying factors to reflect the effects of erosion. The length and location of this applied degradation correspond to field observations of similar blades after approximately two years of operation. Wind tunnel test conducted by Krog Kruse et al.
(2021) utilised P400 and P40 grit sandpaper to simulate distinct erosion levels on a NACA $63_3$-418 aerofoil. These textures, representing rain-induced erosion, provided empirical references for deriving aerofoil polar degradation factors, which were subsequently applied to the aeroelastic model to assess their effects on aerodynamic performance. It is important, however, to acknowledge that real-world degradation of turbine blade leading edges can be influenced by a multitude of factors beyond those captured in this controlled simulation.

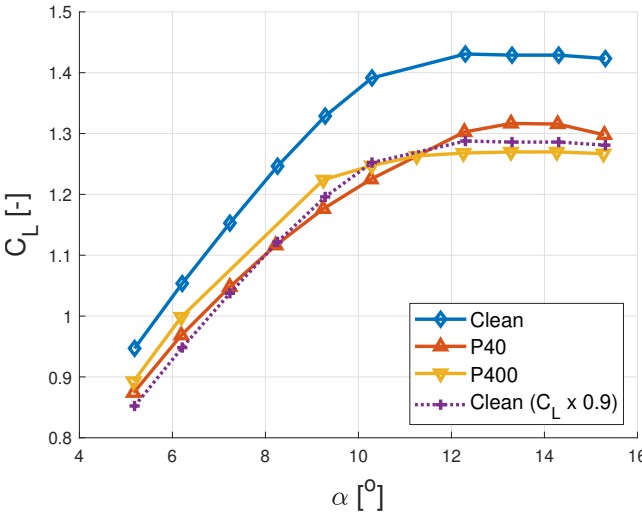 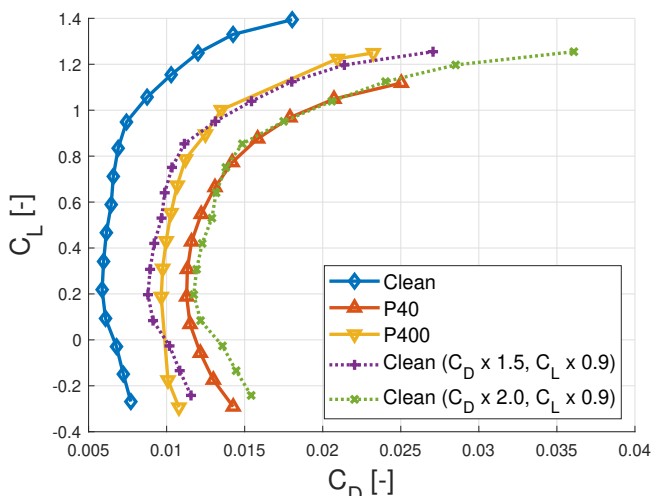

**Figure 1.** Effect of leading-edge erosion on lift coefficient ($C_L$) as a function of angle of attack ($\alpha$). Compares Clean, P40 and P400 blade roughnesses, demonstrating decreased $C_L$ with increased roughness (measurement data from Krog Kruse et al. (2021))

**Figure 2.** Effect of leading-edge erosion on drag coefficient ($C_D$) as a function of lift coefficient $C_L$). Compares Clean, P40 and P400 blade roughnesses, demonstrating increased $C_D$ with increased roughness (measurement data from Krog Kruse et al. (2021))

A challenge in this work was the lack of access to the aerofoil geometry. Despite the aerofoil characteristics being available, they may not be presented due to intellectual property rights. Therefore, the degradation of the proprietary aerofoil characteristics was modelled by applying relative changes derived from wind tunnel tests performed on an alternative aerofoil. While Skrzypinski et al. (2014) proposed a model for altering aerofoil characteristics, this study employed a simplified approach. Al-

though, the tested alternative aerofoil, from which the factors were derived, was not an identical match to that in the multibody model, this method provided a suitable approximation for representing erosion on the outboard region of turbine blades.

Wind tunnel tests on the alternative aerofoil were conducted at a Reynolds number of 5 x $10^6$. Results for the Clean (no sandpaper), P400 (fine, with an average roughness value of 0.035 mm) and P40 (coarse, with an average roughness value of 0.415 mm) sandpapers were used. The P40 sandpaper, which has a larger grain size, was chosen to represent a more severe erosion state. Figures 1 and 2 illustrate that for both P400 and P40 sandpaper roughnesses, the $C_L max$ is reduced by approximately 10% within a specific range of $\alpha$ before deep stall. Similarly, the $C_D$ increased by approximately 50% for P400 roughness and 100% for P40 roughness, compared to a 'clean' aerofoil surface. These percentage changes in lift and drag coefficients were subsequently applied to approximate the degradation of the proprietary aerofoil polars used in the simulation model. For simplicity, the lift polar representing the Clean aerofoil was scaled by a factor of 0.9. Additionally, two artificial drag polars were created by scaling the drag polar representing the Clean aerofoil by factors of 1.5 and 2.0.

This approach was deemed acceptable as the multibody simulations were performed over a limited range of angle of attacks, which are relevant for cases of normal turbine operation, detailed in Section 2.5. These factors were applied between the aerofoil's minimum and maximum lift angles of attack. Beyond this range, at high angles of attack (30 degrees), the adjusted characteristics were smoothly blended into the original data. The assumption being that at high angles of attack the performance is dominated by the flow separation and the resulting pressure distribution, resembles that of a flat plate, thus being less dependent on the specific surface characteristics. Due to confidentiality, the final modified aerofoil characteristics may not be shown.

## 2.3 Representing wind farm turbulence

The simulations reproduced turbulence conditions typical of an operational offshore wind farm. Turbulence data were sourced from a meteorological mast located adjacent to an operational offshore wind farm which utilises the same turbine type as the multibody model.

The turbulence intensity profile at the site, corrected to the turbine's hub height using WindPro EMD International A/S (2023), is shown in Figure 3. This comprehensive dataset was derived from six years of 10-minute averaged data and included all wind speeds without directional filtering. It incorporated the effects of wakes from adjacent turbines as well as a wind farm, offering a realistic depiction of the first row in a wind farm environment.

The mean $TI$ was 7.3% for the entire period and 6.7% when limited to turbine operational wind speeds - between 4 and 25 m/s. The $TI$ distribution is depicted in Figure 4 and together, these figures reveal that although higher turbulence intensities did occur, they were relatively rare and primarily occurred at lower wind speeds. For sake of convenience in the simulation environment, a turbulence intensity of 6% was used to represent mean annual wind farm turbulence with wake free directional filters applied. Specific location details of the wind farm and the met mast are omitted due to confidentiality.

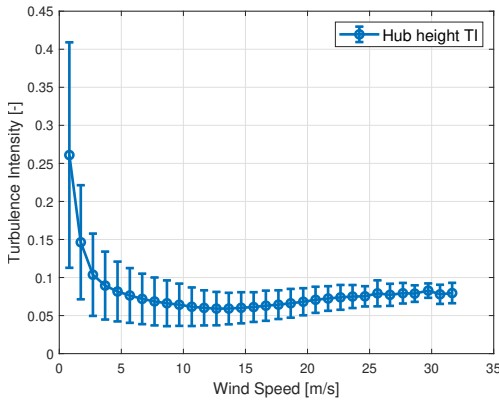 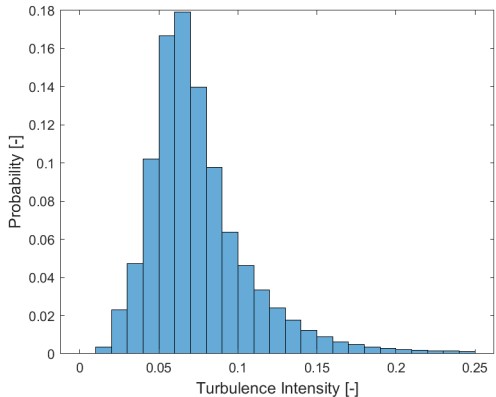

**Figure 3.** Turbulence intensity at the hub height as a function of wind speed. Data obtained from the wind farm's meteorological mast

**Figure 4.** Probability density distribution of turbulence intensity ($TI$) for wind speeds between 4-25 m/s (limited at 25%)

## 2.4 Data time averaging

To better understand the potential impact of different data processing techniques on wind and power measurements, this study investigated the effects of varying time-averaging periods on the detection and quantification of erosion-related power losses. The analysis of wind and power measurements often involves binning and time-averaging. Binning and time-averaging data are forms of data filtering that can both clarify and potentially complicate the interpretation of results. Careful selection of bin sizes is crucial to avoid information loss and potential misinterpretation.

Data time averaging, traditionally over a 10-minute period, is used to smooth turbine signals such as wind speed, power and behaviours such as pitch or torque. These responses are slightly delayed to wind speed, that can fluctuate rapidly. Time averaging can provide a more representative overview of turbine performance and prevailing wind conditions, allowing identification of trends, patterns in data, supported by findings from Abolude and Zhou (2018), Do and Berthaut-Gerentes (2018) and Elliott and Infield that express associated benefits and complexities. While longer periods simplify data processing and reduce data storage needs, they also risk masking changes in performance and the subtle effects of leading-edge erosion on turbine dynamics (Gonzalez et al. (2017); Gonzalez et al. (2019)).

Importantly, time averaging potentially introduces bias into data analysis. For instance, smoothing out short-term fluctuations in power output can inadvertently alter the perceived shape of the power curve, such as the location of the knee in the power curve. A crucial aspect to consider is the balance between the need to reduce noise in the data and the risk of masking important turbine responses. An excessively short time period may lead to noisy data, whereas a period that is too long risks over-filtering the turbine's behaviour.

Furthermore, time averaging affects the perceived inertia of the turbine. When power output is averaged over a longer time period, short-term fluctuations in power output are suppressed, potentially making the turbine appear less responsive to changes in wind speed. If the time period used for averaging significantly exceeds the characteristic response time of

the turbine, the inertia of the turbine may be underestimated and its ability to respond to changes in wind speed could be overestimated. Conversely, using a time period that is too short may amplify short-term fluctuations in power output, making data interpretation difficult because the raw data, in many cases, shall be a swarm of data points. It is therefore important that the specific requirements of the analysis should ultimately dictate the selected time averaging period.

To investigate these effects, this study explored the use of shorter time-averaging periods to potentially unravel the nuanced effects of leading-edge erosion on turbine performance, which may be masked in traditional 10-minute averages. The challenge lies in selecting an period that offers sufficient detail without sacrificing clarity, ensuring that critical information about turbine performance and the impact of blade surface conditions is neither lost nor misrepresented. Data from multibody simulations, with a 0.01 second time step, were collected from all wind speed simulation seeds for a given turbulence intensity and blade

profile. Time averaging was then applied to wind speed and turbine sensor variables such as power for time periods of 0.01, 1, 30, 60, 120, 300 and 600 seconds. Subsequently, the data were averaged into 1 m/s wind speed bins and the turbulence intensity of the original simulation seed was applied to time periods sliced from it.

## 2.5    Simulation settings and test cases

This study employed a range of simulation cases using HAWC2, a Blade Element Momentum (BEM)-based multi-body aero-

servo-elastic tool, to explore the impact of turbulence intensity and blade erosion on wind turbine performance. Simulations were executed for a range of turbulence intensities for the Clean and and two eroded blade profiles. Individual cases were run in 1 m/s increments ranging from 4 to 25 m/s, representing the turbine's cut-in and cut-out wind speeds. Each configuration of wind speed, $TI$ and blade condition was represented by six individual simulation runs, or seeds, to ensure statistical robustness as per the International Electrotechnical Commission (IEC) (2019) 61400-1 standard.

The turbulence intensity was varied across a broad spectrum including 0%, 4%, 5.5%, 6.0%, 6.5%, 7%, 10%, 15% and 20%, with a focus on values around the observed average annual ambient $TI$ at an offshore site, along with broader values for comparison. Each simulation was run for 900 seconds, with data from the last 600 seconds used for analysis to ensure steady-state conditions had been reached. The time step of the simulations was 0.01 seconds. The wind shear was investigated for two conditions, including a zero shear value and a power-law profile with an alpha value of 0.14. The air density was fixed

at 1.225 kg/m$^3$, representative of sea-level conditions at 15$^o$C. The Mann turbulence parameter $\alpha\epsilon^{2/3}$ Mann (1994), energy level was set to its default value of 1.0. For a detailed explanation of specific parameters and settings, refer to the HAWC2 manual Larsen and Hansen (2007) or International Electrotechnical Commission (IEC) (2019) 61400-1 standard.

## 3    Results and discussion

The simulations conducted in this study were analysed from multiple perspectives, with the results presented in four distinct

sections:

- – Effect of shear and blade erosion on power

– Effect of turbulence intensity and blade erosion on power

– Effect on annual energy production (AEP)

– Effect of erosion, time averaging and turbulence on power curve

## 3.1 Effect of shear and blade erosion on power

This section examines the impact of leading edge erosion on wind turbine power curves under different wind shear conditions using multibody simulations. The simulations were executed at a constant turbulence intensity of 0% to isolate the distinct effects of shear and blade condition. Figure 5 presents normalised power curves for Clean blades and those exhibiting P400 and P40 roughness levels, under both zero shear and with imposed wind shear conditions of a power-law profile with an alpha

value of 0.14 . As expected, the leading-edge roughness reduced the power output across the range of wind speeds.

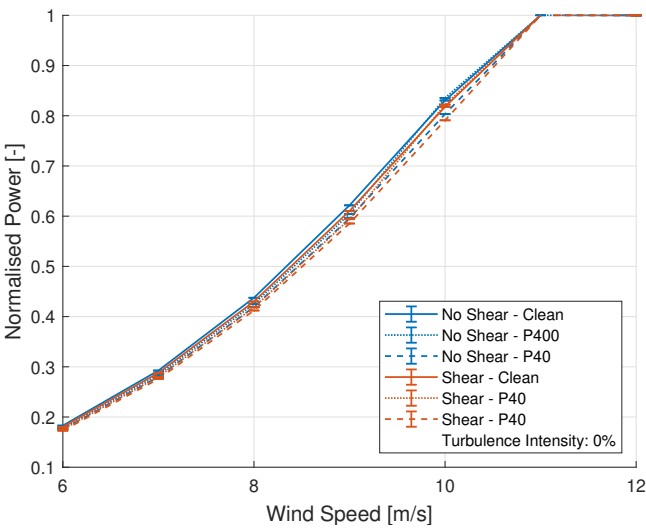

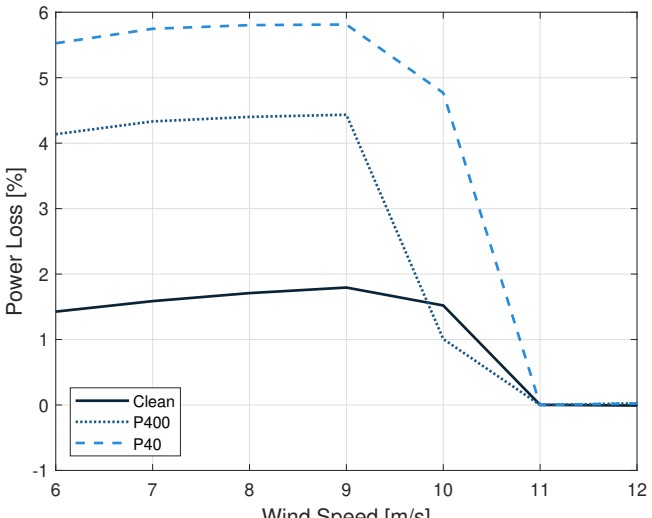

**Figure 5.** Effect of various blade conditions compared to that of shear and no-shear wind condition on the power curve (0% $TI$)

**Figure 6.** Percentage power loss due to shear, referenced against the baseline Clean blade without shear, for various blade conditions (0% $TI$)

Comparing the no-shear and shear conditions revealed the turbine's sensitivity to shear-induced variations in the wind profile along the rotor span. Under shear conditions, the power curves for both Clean and eroded blades exhibited a shift, up to 5.8% for the P40 roughness blade with shear, relative to a Clean blade at zero shear conditions, as seen in Figure 6. This demonstrates an adjustment in operational behaviour to account for the velocity gradient imposed by the atmospheric shear and the convoluting

effects on power of shear.

Despite these observed shear effects complicating the isolation of variables and highlighting the difficulty of analysing real-world measurement data, this analysis focuses on investigating turbulence, as it is an atmospheric condition whose impact on

performance is typically more substantial than that of wind shear (Saint-Drenan et al. (2020)). Although wind shear remains relevant, the intention is not to investigate each atmospheric condition in detail but rather to illustrate the effects through select examples.

## 3.2 Effect of turbulence intensity and blade erosion on power

### 3.2.1 Investigation based on the power curves

The normalised, 10-minute averaged power curve of the turbine for various turbulence intensities is shown in Figure 7. Consistent with previous research Saint-Drenan et al. (2020), Wagner et al. (2010), the turbine's power output is significantly influenced by turbulence intensity ($TI$), particularly within the partial load region of the power curve, which represents the operational range between the wind speed where maximum rotational speed is achieved and the wind speed where rated power is reached. The plot includes higher turbulence intensities, such as 20%, to demonstrate the trend in their effect on the power curve. This variation expresses the considerable effect of turbulence intensity on turbine performance.

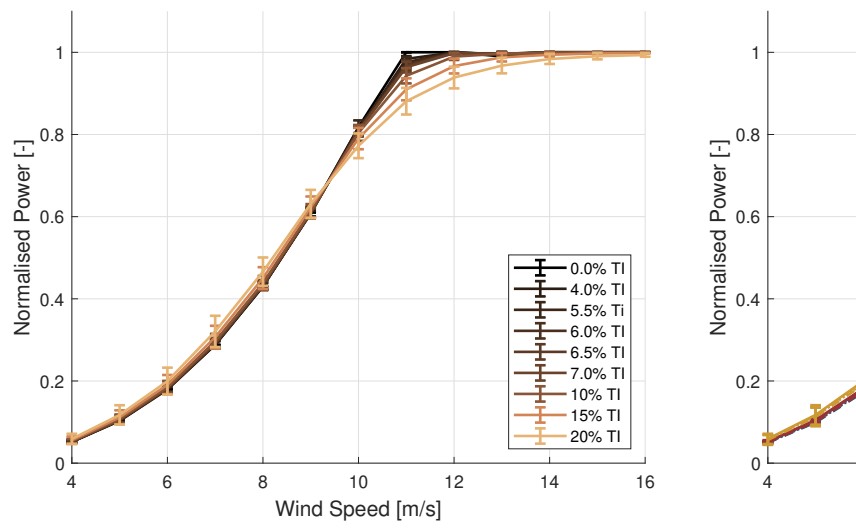

**Figure 7.** Effect of turbulence intensity on the power curve (Clean blades)

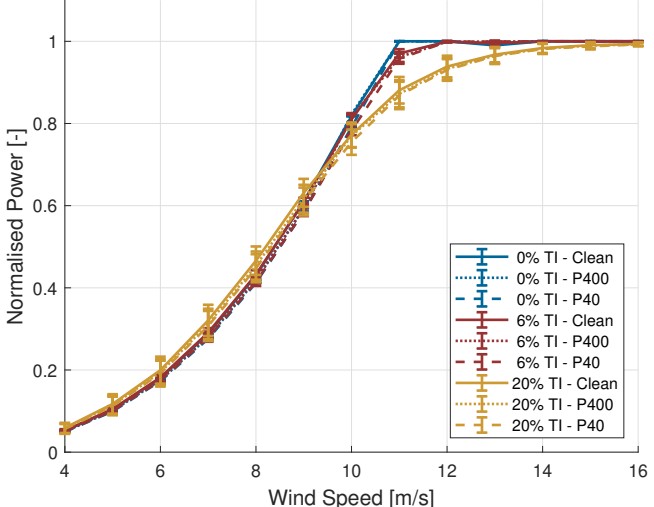

**Figure 8.** Effect of three specific turbulence intensities compared to that of three blade profiles on the power curve

Comparative analysis among Clean, P400, and P40 blade conditions, representing varying degrees of erosion applied on the leading edge of the last 15% of the blade length, is presented in Figure 8. Results are shown for 6% turbulence intensity, representing a typical mean value for offshore sites. The 0% and 20% plots are included for comparison to more outlying conditions, demonstrating a similar trend in power reduction with increasing blade erosion. Similar effect on omitted power curves affirms the consistent detrimental impact of erosion across various $TI$ conditions.

The figures facilitate a revealing comparison of the relative effects of turbulence and erosion. Analysis of the power curve at specific points, such as the 'knee', reveals that changes in turbulence intensity influence power output more significantly than

blade erosion. This is evident in Figure 8: for the Clean blade at 11 m/s wind speed, power reduces to approximately 97.0% when $TI$ increases from 0% to 6% and further to 88.1% at 20% $TI$. For eroded blades, these reductions are comparable: 96.2% and 87.2% (P400) and 95.7% and 86.9% (P40).

Considering a wind speed of 11 m/s and 6% $TI$, erosion causes power losses of approximately 0.9% (P400) and 1.3% (P40) relative to the Clean blade. Importantly, the power output's standard deviation at this wind speed is approximately 1.03% (6% $TI$) and 3.23% (20% $TI$). This indicates a major challenge: particularly at higher $TI$, the standard deviation exceeds the power loss due to roughness, making it difficult to isolate and detect the effects of erosion on power output based on the power curve alone. Yet, the comparability of values at lower $TI$ suggests that erosion effects could potentially be detected more readily under less turbulent conditions.

An interesting observation in Figure 8 is the intersection of power curves around 9.5 m/s. This intersection is caused by a combination of factors. Firstly, the inflection point in the power curve at 9.5 m/s, where the curvature changes, plays a role. Secondly, the averaging effects inherent in calculating power curves from unsteady power output contribute to this phenomenon.

While analysing the changes in power curve shapes provides valuable insights, it offers an incomplete understanding of the true impact of erosion and turbulence. To accurately assess the overall effect, it is crucial to consider the site-specific wind speed distribution and its influence on the turbine's annual energy production. A more comprehensive analysis is presented in Section 3.3.

### 3.2.2 Investigation relative to a reference power curve

To further investigate how the power curve is influenced by erosion under varying turbulence intensities, this study conducted a comparative analysis. The change in power relative to a reference Clean profile power curve at 6% $TI$, focusing on P40 roughness, was investigated. The results are shown in Figure 9 as a function of wind speed across a range of turbulence intensities. The delta power curve exhibits a 'kink', a point characterised by a sudden change in gradient, at around 9.5 m/s attributed to the previously discussed effect of time averaging. The most substantial reduction of power due to roughness were identified between 9 and 13 m/s. At lower turbulence intensities, i.e. 7% and below, roughness was found to have an effect in reducing power. Moreover, for increasing turbulence intensities, the influence of roughness increased dramatically within the same wind speed range.

These findings highlight the non-linear and interdependent relationship between blade roughness and turbulence intensity in their impact on power output. Furthermore, they suggest that both factors must be considered when assessing wind turbine performance, especially within specific wind speed ranges.

### 3.2.3 Investigation using power coefficients

The coefficient of power ($C_p$) represents a key metric for evaluating the performance of wind turbines. This study analysed how $C_p$ varies with wind speed, turbulence intensity and blade roughness. The rationale for investigating $C_p$ was based on the understanding that turbulence intensity does not inherently alter the efficiency of the wind turbine; rather, it is the combination

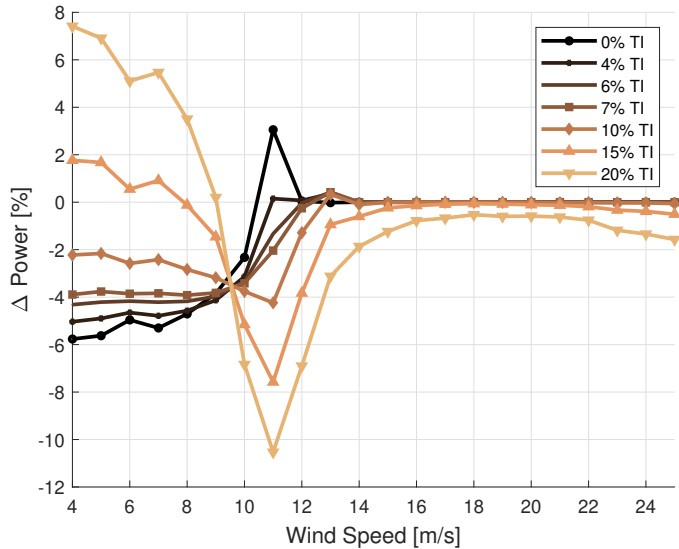

**Figure 9.** P40 profile - Percentage change in power output from the Clean baseline as a function of wind speed, showing impact of roughness and $TI$ (Baseline: Clean profile, $6\%$ $TI$)

of turbulence intensity and the time averaging period that can lead to erroneous conclusions. The power coefficient is calculated using the equation:

$$C_p = \frac{P}{0.5 \cdot \rho \cdot V^3 \cdot \pi \cdot R^2} \tag{1}$$

where $P$ is the power, $\rho$ is the air density, $V$ is the wind speed and $R$ is the rotor radius. Here, $C_p$ is computed based on the averaged values of wind speed and power. It is important to note that the averaging was performed on wind speed and power separately before calculating $C_p$. This investigation also highlighted the contrast between steady-state aerodynamic analysis with zero turbulence intensity and analysis that includes turbulence intensity. Figure 10 shows the $C_p$ as a function of wind speed for various turbulence intensities, employing a Clean profile blade. The findings indicate that the greatest variation of $C_p$ is observed at wind speeds below approximately 9 m/s. To evaluate the impact of roughened blade leading edges on $C_p$, Figure 11 shows the variation of $C_p$ for the profiles at 6% turbulence intensity. These results suggest that the impact of both forms of roughness is less pronounced than that of a certain threshold value of turbulence intensity.

This investigation analysed multibody simulated data, focusing on the last 10 minutes of each simulation to capture steady-state conditions. Instances where the power coefficient ($C_p$) exceeded or approached the Betz limit of 0.593 in high turbulence intensity conditions were carefully examined. The exceeding of the Betz limit may be attributed to several factors, including turbine inertia and control dynamics, where the inherent latency in response mechanisms such as pitch and generator torque control results in a temporal mismatch between the turbine's power response and rapid wind speed fluctuations characteristic of turbulent environments. This mismatch, particularly when results are time-averaged over a 10-minute window, can yield

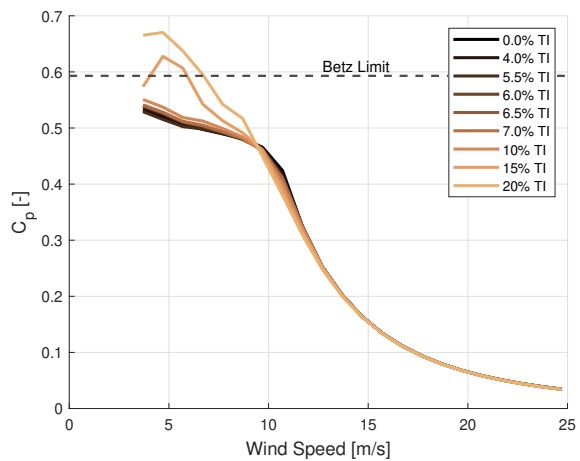 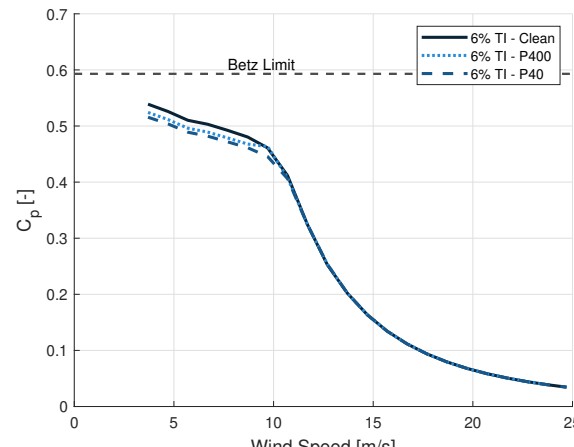

**Figure 10.** Power coefficient as a function of wind speed for a Clean profile blade, with various turbulence intensities

**Figure 11.** Power coefficient as a function of wind speed for three leading edge profiles (6% $TI$)

simulated $C_p$ values that in some conditions surpass the Betz limit. Thus, it is believed that $C_p$ values exceeding the Betz limit have no physical meaning, rather they are an artefact from the averaging of the wind speed and the rotor performance. Therefore, the analysis can lead to erroneous conclusions.

Additionally, the analysis revealed that highly turbulent conditions create localised gusts, temporarily increasing the effective wind speed at segments of the rotor, diverging from steady-state assumptions and causing transient spikes in power output, further exacerbating the mismatch between wind speed and power output. This effect, coupled with the stochastic nature of turbulence that can enhance kinetic energy transfer to the rotor plane and momentarily boost the available wind energy beyond typical averages used in Betz limit calculations. These findings underscore the limitations of steady-state assumptions in accurately capturing the dynamic interactions between wind turbines and complex wind fields. Future research efforts should focus on refined simulation models and analysis techniques designed to address these limitations.

Figure 12 provides further insight on the combined effects of roughness and turbulence intensity. It depicts $C_p$ for a limited range of lower turbulence intensities, along with the three blade profiles at 6% $TI$ for wind speeds up to 11 m/s. The overlap between the $C_p$'s for turbulence intensity and roughness suggests that distinguishing between these two effects may be challenging due to the 'masking' effect, particularly in high turbulence conditions. This complicates the interpretation of aerodynamic performance degradation caused by blade erosion.

### 3.2.4 Summary of the influence of $TI$ and erosion on power

The findings presented herein reinforce the notion that both turbulence and blade erosion exert substantial influences on the wind turbine power output. It has been observed that turbulence profoundly affects the power curve, predominantly in the partial load region. Despite the inherent complexities associated with analysing the performance of the wind turbines under

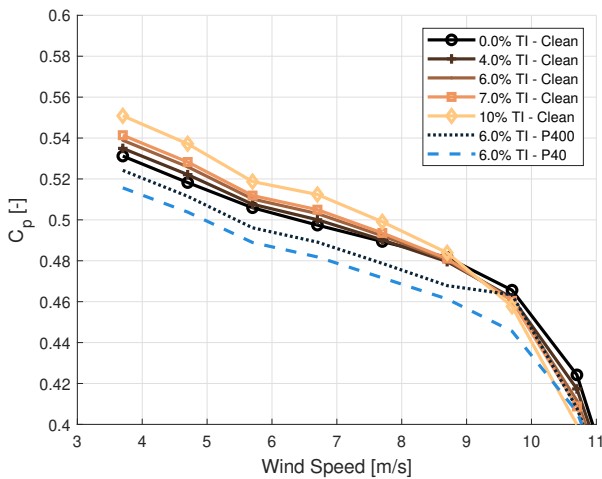

**Figure 12.** Power coefficient as a function of wind speed for a Clean profile blade at various turbulence intensities and various leading edge roughness profiles at 6% $TI$

turbulent conditions, this study emphasises the significance of incorporating $TI$ in performance evaluations. This alignment
with preceding studies Wagner et al. (2010) and Saint-Drenan et al. (2020) further validates the critical nature of $TI$ in such analyses.

The examination of delta power shows the detrimental effects blade erosion has on wind turbine power output, with the greatest power reduction due to roughness observed at wind speeds between 9 and 13 m/s. This observation is consistent with prior research Bak et al. (2020), emphasising the significance of considering roughness effects when assessing wind turbine
performance. The study also showed that the impact of roughness on power output is further exacerbated at higher turbulence intensities, suggesting that both turbulence and erosion should be considered in performance assessment.

While the analysis focused on the impact of blade erosion on power, it is important to recognise that erosion could also influence other aspects such as loads and sensor output. These potential impacts warrant further investigation.

### 3.3 Annual energy production (AEP) calculation

This section explores the calculation of annual energy production, investigating the impact of both blade erosion and turbulence on wind turbine performance. Analyses included a real-world operational offshore wind farm and hypothetical scenarios at three fictitious sites.

### 3.3.1 AEP for an existing site

AEP was calculated for a wind turbine situated in an offshore wind farm operating under mean turbulence intensity of 6%, characterized by a Weibull distribution with a scale parameter $A = 10.72$, a shape parameter of $k = 2.17$. This corresponds to an average wind speed of 9.49 m/s. The computations expressly excluded the wake effects of upstream wind turbines.

The comparative analysis focused on quantifying the impact of blade erosion and turbulence intensity on AEP by comparing the outcomes for three distinct blade profiles. Table 1 shows the AEP variation for each profile relative to the 6% $TI$ power curve of the corresponding profile.

**Table 1.** Change in AEP as a function of $TI$. Row 2 shows AEP change relative to "Clean" performance at $TI$=6%; Row 3 shows AEP change relative to "P400" performance at $TI$=6%; Row 4 shows AEP change relative to "P40" performance at $TI$=6% with $V_{ave}$=9.49 m/s

| Blade profile | TI [%] | | | | | | | |
|---|---|---|---|---|---|---|---|---|
| | 0 | 4 | 5.5 | 6 | 6.5 | 7 | 10 | 15 |
| Clean delta AEP [%] | 0.34 | 0.11 | 0.03 | 0 | -0.02 | -0.04 | -0.23 | -0.63 |
| P400 delta AEP [%] | 0.44 | 0.11 | 0.05 | 0 | -0.04 | -0.07 | -0.29 | -0.78 |
| P40 delta AEP [%] | 0.51 | 0.12 | 0.05 | 0 | -0.04 | -0.06 | -0.26 | -0.70 |

**Table 2.** Change in AEP as a function of $TI$ and roughness level: AEP change relative to "Clean" performance at $TI$=6% with $V_{ave}$=9.49 m/s

| Blade profile | TI [%] | | | | | | | |
|---|---|---|---|---|---|---|---|---|
| | 0 | 4 | 5.5 | 6 | 6.5 | 7 | 10 | 15 |
| Clean delta AEP [%] | 0.34 | 0.11 | 0.03 | 0 | -0.02 | -0.04 | -0.23 | -0.63 |
| P400 delta AEP [%] | -0.38 | -0.71 | -0.77 | -0.82 | -0.86 | -0.89 | -1.10 | -1.59 |
| P40 delta AEP [%] | -0.96 | -1.33 | -1.41 | -1.46 | -1.49 | -1.51 | -1.71 | -2.14 |

Similarly, Table 2 shows the AEP variation for each profile relative to the Clean blade profile's 6% $TI$ power curve. From the results it is clear that even mild simulated erosion, represented by the P400 blade profile, had a significant impact on the turbine's AEP, with a 0.82% decrease. As erosion progressed, the AEP decreased further to 1.46% for the rougher P40 sandpaper, relative to a Clean blade. Moreover, once a blade is rough, its impact on AEP relative to the Clean blade profile is significant.

Table 2 also presents turbulence intensities impact on AEP. As turbulence intensity increased, the AEP decreased for all blade profiles. The impact was more significant for the rougher blade profiles, with the P40 sandpaper profile already showing a high decrease in AEP at 2.14% for 15% turbulence intensity.

### 3.3.2 AEP for three fictitious sites with varying wind speeds

The investigation extended AEP calculations to three hypothetical sites, each characterised by average wind speeds of 6, 8 and
10 m/s. The subsequent AEP variations for each blade profile, relative to the Clean blade profile's 6% $TI$ power curve, are presented in Table 3 for an average wind speed of 6 m/s, Table 4 for an average wind speed of 8 m/s and Table 5 for an average wind speed of 10 m/s. Three different climates were investigated:

- 6 m/s average wind speed: k = 2, A= 6.8 m/s (Table 3)

- 8 m/s average wind speed: k = 2, A= 9 8 m/s (Table 4)

330  - 10 m/s average wind speed: k = 2, A= 11.3 m/s (Table 5)

**Table 3.** Change in AEP as a function of $TI$ and roughness level: AEP change relative to "Clean" performance at $TI$=6 % with $V_{ave}$=6 m/s

| Blade profile | TI [%] | | | | | | | |
|---|---|---|---|---|---|---|---|---|
| | 0 | 4 | 5.5 | 6 | 6.5 | 7 | 10 | 15 |
| Clean delta AEP [%] | 0.16 | -0.04 | -0.20 | 0 | 0.02 | 0.05 | 0.25 | 0.86 |
| P400 delta AEP [%] | -1.20 | -1.49 | -1.65 | -1.47 | -1.46 | -1.44 | -1.28 | -0.76 |
| P40 delta AEP [%] | -2.51 | -2.84 | -3.02 | -2.83 | -2.82 | -2.79 | -2.60 | -2.00 |

**Table 4.** Change in AEP as a function of $TI$ and roughness level: AEP change relative to "Clean" performance at $TI$=6 % with $V_{ave}$=8 m/s

| Blade profile | TI [%] | | | | | | | |
|---|---|---|---|---|---|---|---|---|
| | 0 | 4 | 5.5 | 6 | 6.5 | 7 | 10 | 15 |
| Clean delta AEP [%] | 0.32 | 0.08 | -0.02 | 0 | -0.01 | -0.03 | -0.13 | -0.31 |
| P400 delta AEP [%] | -0.51 | -0.85 | -0.94 | -0.94 | -0.97 | -1 | -1.13 | -1.40 |
| P40 delta AEP [%] | -1.39 | -1.78 | -1.89 | -1.88 | -1.91 | -1.92 | -2.03 | -2.24 |

**Table 5.** Change in AEP as a function of $TI$ and roughness level: AEP change relative to "Clean" performance at $TI$=6 % with $V_{ave}$=10 m/s

| Blade profile | TI [%] | | | | | | | |
|---|---|---|---|---|---|---|---|---|
| | 0 | 4 | 5.5 | 6 | 6.5 | 7 | 10 | 15 |
| Clean delta AEP [%] | 0.30 | 0.10 | 0.03 | 0 | -0.02 | -0.04 | -0.21 | -0.61 |
| P400 delta AEP [%] | -0.67 | -0.96 | -1.02 | -1.06 | -1.10 | -1.13 | -1.32 | -1.80 |
| P40 delta AEP [%] | -0.84 | -1.18 | -1.25 | -1.29 | -1.32 | -1.34 | -1.52 | -1.95 |

From these results it may be concluded that the impact of turbulence intensity on AEP is more pronounced at lower average wind speeds. This observation is evidenced by the more substantial AEP reductions observed at lower $TI$ levels for the P400

and P40 blade profiles, as well as the higher AEP decrease at higher $TI$ levels for the Clean blade profile, at lower average wind speeds.

Simultaneously it is obvious, that the impact of blade erosion on AEP is more significant for lower average wind speeds. This is evident from the larger AEP decrease due to blade erosion for the P400 and P40 blade profiles, as well as the higher AEP decrease for the Clean blade profile, at higher average wind speeds.

The large loss due to erosion for $V_{ave}$=6 m/s is due to the fact that much of the energy is produced below rated power, where erosion has a significant impact. Erosion has almost no impact at rated power. Smaller losses due to erosion are seen for $V_{ave}$=10m/s. The higher the $TI$, the greater the gain when most of the production occurs at low wind speeds, as power increases below 9.5 m/s due to the averaging. Conversely, the higher the $TI$, the greater the loss when most of the production occurs at high wind speeds, as the power decreases above 9.5 m/s.

Also a trend emerges, suggesting that the comparative effects of blade erosion and turbulence intensity on AEP vary contingent upon the average wind speed and the specific blade profile under consideration. For instance, at an average wind speed of 6 m/s, blade erosion has a larger impact on AEP than turbulence intensity for all blade profiles. At higher wind speeds, turbulence intensity has a more pronounced impact on AEP, particularly evident in the context of the P40 blade profile.

### 3.3.3   Summary of the effect of $TI$ and erosion on AEP

The investigation into Annual Energy Production encompassed both:

  – A specific actual wind climate

  – Three artificial wind climates

For the first AEP calculation the AEP variation for the three blade profiles pertaining to a specific climate with a mean wind speed of 9.49 m/s revealed that even minimal simulated erosion, represented by the P400 blade profile, could precipitate a notable reduction in AEP by 0.82%. As erosion progressed, the AEP decreased further to 1.46% for the coarser P40 sandpaper, relative to a Clean blade. Furthermore, the effect of a blade's roughness on AEP in comparison to the Clean blade profile was substantial.

The second study additionally examined how three different site specific mean average wind speeds (6, 8 and 10 m/s) affected AEP for the three blade profiles. The findings indicate that at lower wind speeds, the AEP variation caused by turbulence intensity in comparison to a Clean blade profile is more important. This result underlines the importance of considering the level of turbulence intensity and its impact on AEP in wind farm site selection and design considerations. Notably, the findings from the hypothetical scenario with the highest wind speed at 10 m/s corresponded well to the first AEP calculation for the specific wind climate.

From the study it was observed that alterations in $TI$ invariably influence AEP. Such variability introduces complexities in accurately attributing changes in AEP solely to erosion, as fluctuations in $TI$ could equally account for observed variations.

### 3.4 Influence of erosion, data time averaging and turbulence intensity on the power curve

This section examines how blade erosion, data time averaging periods and turbulence intensity affect wind turbine power curves. Simulations were conducted employing both Clean and eroded (P40 roughness) blade profiles.

**Impact of time averaging periods from a baseline of 0.01 s time period at 15% $TI$**

Figure 13 illustrates the power as a function of wind speed for different time averaging periods at a fixed turbulence intensity of 15%. This fixed turbulence intensity was chosen as the baseline to demonstrate solely the impact of time averaging periods

at various wind speeds on the power output. The graph focuses on both the low speed region and the knee of the power curve to showcase the varied impact of time averaging across different wind speeds. To quantify these effects Figure 14 presents the percentage change in power relative to the baseline case (Clean profile, 0.01-second period, fixed 15% $TI$). This visualisation demonstrates the deviations in power output across various averaging periods, especially at lower wind speeds. By using a fixed turbulence intensity as the baseline, the 15% $TI$ example demonstrated significant reductions in observed power with

longer averaging periods, with smaller time periods showing lower deviations. Notably, the 1-second period exhibited a more neutral impact on power deviation across the range of wind speeds.

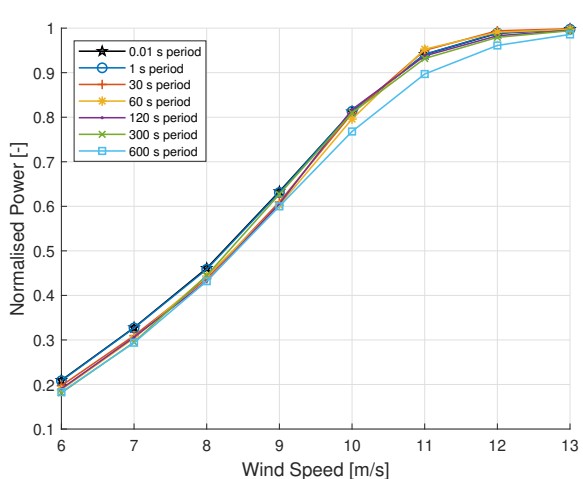

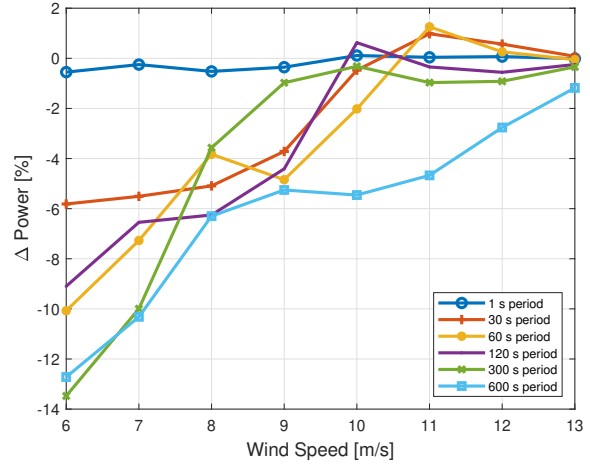

**Figure 14.** Clean profile - Percentage change in power output as a
**Figure 13.** Clean profile - Normalised power as a function of wind function of wind speed for multiple time averaging periods, showing
speed for multiple time averaging periods, showing impact of time impact of time periods (Baseline: Clean profile, 0.01 s period 15%
periods (Baseline: Clean profile, 0.01 s period, 15% $TI$) $TI$)

**Impact of time averaging periods from a baseline of 0.01 s time period with matched $TI$**

To further investigate time averaging's effects, a baseline case with a Clean profile and a 0.01-second period was used, with the turbulence intensity of the baseline adjusted to match that of the analysed point. The percentage difference in power from

380 the baseline case was calculated for various turbulence intensities for a set of time averaging periods as illustrated in Figure 15,

showing the results for a fixed wind speed of 7 m/s, representing the low speed region of the power curve. The data revealed the impact of time periods as:

– The 1-second time period showing only a marginal effect

– A trend of increasing power reduction with increasing turbulence intensity

– Larger time periods resulting in greater percentage decreases in power

In contrast, Figure 16 presents the results for a fixed wind speed of 11 m/s, which corresponds to the knee of the power curve. The following observations are made:

– Different time periods exhibiting both increasing and decreasing effects on power output

– Lower time periods (30 and 60 seconds) pulling the power curve upward

390     – 1-second and 120-second periods having a more neutral effect

– Increasing turbulence intensity having a somewhat linear influence on power change

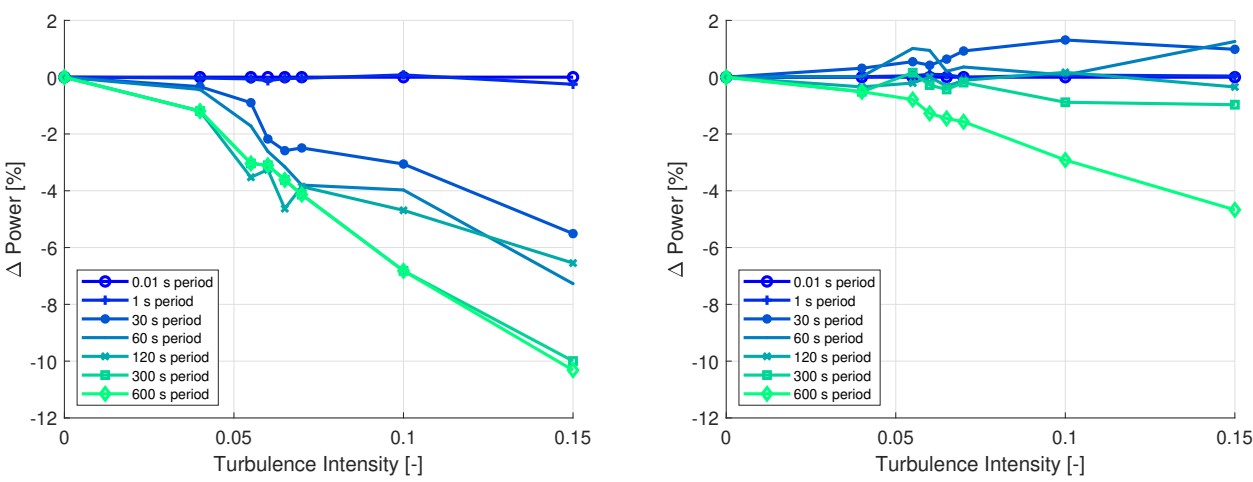

**Figure 15.** 7 m/s, Clean profile - Percentage change in power output from the 0.01 second baseline as a function of turbulence intensity, showing impact of time periods (Baseline: Clean profile, 0.01 s period, matched $TI$)

**Figure 16.** 11 m/s, Clean profile - Percentage change in power output from the 0.01 second baseline as a function of turbulence intensity, showing impact of time periods (Baseline: Clean profile, 0.01 s period, matched $TI$)

In both cases, longer time periods generally showed higher delta power than shorter periods. Still, the magnitude of the change appeared larger for the 7 m/s wind speed, indicating a potentially greater impact of time periods on power output for lower wind speeds. These findings highlight the complex interaction between time averaging periods at various turbulence

intensities for two wind speeds, reinforcing the importance of considering these factors when analysing wind turbine power performance.

**Impact of erosion, time averaging and turbulence from a baseline of a Clean blade and fixed 0% $TI$**

To assess the combined effects of time averaging and turbulence as well as to compare their impacts, initially a Clean blade (i.e. no erosion) profile's impact on power was analysed from a baseline case of a Clean profile at 0.01-second period and, unlike in the previous two cases, with a fixed 0% TI. The impact on power for various time periods at various turbulence intensities is presented in Figures 17 and 18. Note that erosion was not yet considered. It was observed that:

- Again, the 1 second time period has a minimal distorting affect for all turbulence intensities and both wind speeds

- At 7 m/s the effect of 15% $TI$, is up to 18% increase in power for the 0.01 and 1 second time periods

- At 7 m/s looking at the combined effect of time averaging and turbulence, at 15% $TI$ the effects have a 6.5% power increasing effect for the 600 second time period

- At 11 m/s the effect of 15% $TI$, is an up to approximately 4.7% decrease in power for the 0.01 and 1 second time periods

- At 11 m/s looking at the combined effect of time averaging and turbulence, at 15% $TI$ the effects have a approximately 9% power decreasing effect for the 600 second time period

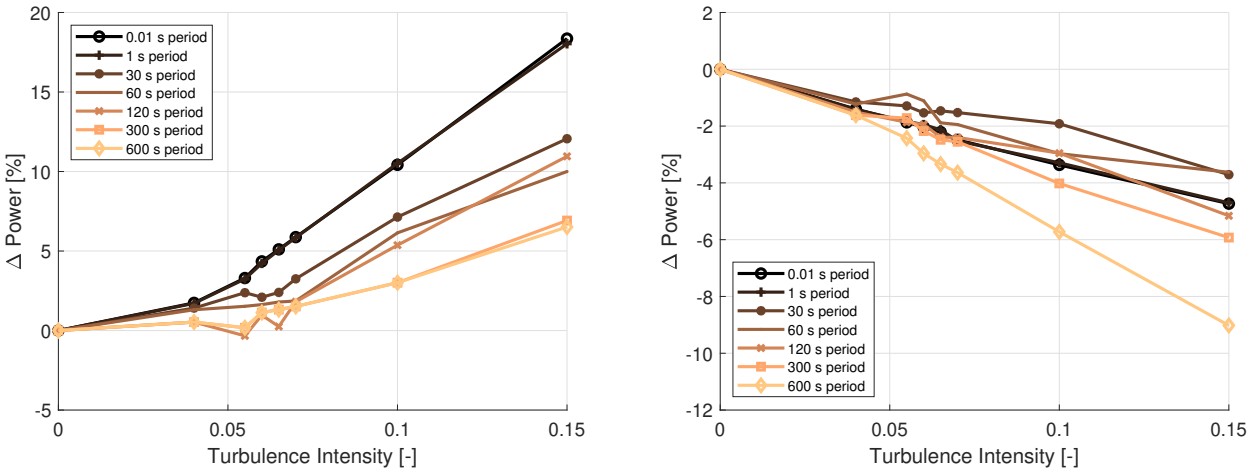

**Figure 17.** 7 m/s, Clean profile - Percentage change in power output from the 0.01 second baseline as a function of turbulence intensity, showing impact of time averaging and $TI$ (Baseline: Clean profile, 0.01 s period, 0% $TI$)

**Figure 18.** 11 m/s, Clean profile - Percentage change in power output from the 0.01 second baseline as a function of turbulence intensity, showing impact of time averaging and $TI$ (Baseline: Clean profile, 0.01 s period, 0% $TI$)

Adding the dimension of blade erosion, which is of particular importance to this study, represented by a P40 roughness, Figures 19 and 20 display results for erosion's influence in addition to time averaging and turbulence. The baseline remained the Clean blade with 0.01-second period and a fixed 0% TI. With the additional aspect of erosion it was observed that:

- Yet again, the 1 second time period has a minimal distorting affect for all turbulence intensities and both wind speeds, despite blade erosion

- At 7 m/s the erosion, in general, reduces the power across all turbulence intensities with an approximately 4% power reduction is observed at 0% turbulence intensity. And with a shift when comparing to Figure 17.

- At 11 m/s erosion's effect seems milder impacting power less dramatically than at the lower wind speed

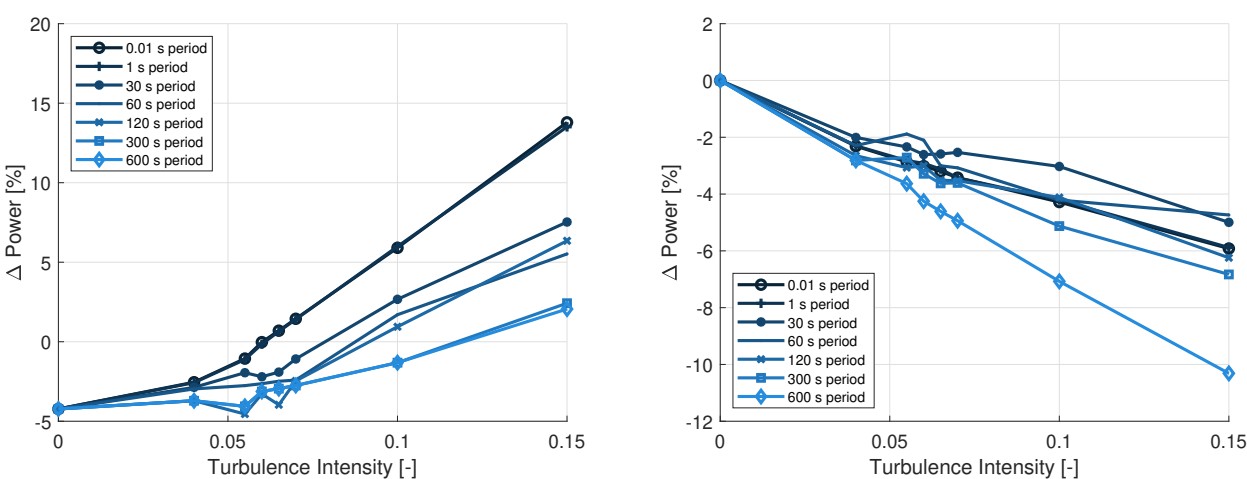

**Figure 19.** 7 m/s, P40 profile - Percentage change in power output from the 0.01 second baseline as a function of turbulence intensity, showing impact of roughness, time averaging and $TI$ (Baseline: Clean profile, 0.01 s period, 0% $TI$)

**Figure 20.** 11 m/s, P40 profile - Percentage change in power output from the 0.01 second baseline as a function of turbulence intensity, showing impact of roughness, time averaging and $TI$ (Baseline: Clean profile, 0.01 s period, 0% $TI$)

These findings emphasise the importance of choosing appropriate time periods for data analysis. Short periods can introduce noise, while long periods can mask important turbine behaviour. The 1-second period balances reducing variability without losing significant information. It is crucial to note that the effect of time averaging on power output varies with wind speed and turbulence intensity, precluding a universal correction. For accurate correction, it would be important to use both a turbine simulation model and meteorological mast data for precise $TI$ measurements when correcting for time averaging influences.

### 3.4.1 Summary of the influence of time averaging on power curve

The investigation into time averaging effects on power analysis showed the significant impact of time period selection on the resulting power curve. Simulation outcomes revealed that larger time periods generally lead to a more pronounced decrease

in power output with increasing turbulence intensity. Despite this, the impact of time averaging on power output varies with operational conditions. At lower wind speeds, larger periods result in a significant power decrease, while at higher wind speeds, smaller periods can increase power output and larger time periods can decrease it. Notably, a 1-second time period maintained a neutral effect across all turbulence intensities.

Comparing the P40 roughness blade to a Clean blade at 0% turbulence intensity demonstrated that the blade surface rough-
ness's impact on power output is less pronounced than time averaging, although both factors significantly affect the power curve. Time averaging can obscure changes in wind turbine performance due to subtle aerodynamic efficiency modifications, such as blade erosion. Short-term changes are harder to detect because averaging smooths out fluctuations in the turbine's response to changes in wind speed and other variables.

To address this issue, selecting shorter averaging periods is advisable to capture transient variations in turbine performance.
Although shorter periods may produce noisier data, this trade-off is necessary for detailed analysis. The study discerned minimal information loss with 1 second values and generally, shorter periods led to smaller losses. While simulations provide good signal control, applying short averaging periods to measured data presents additional challenges due to greater uncertainties in real-world measurements.

It may be argued that the standard deviation of average values can compensate for the effect of time averaging. The standard
deviation of average values can partially offset time averaging effects by estimating lost short-term variability. Nevertheless, if the averaging period exceeds sensor response times significantly, this loss cannot be fully compensated by standard deviation calculations.

### 3.5   Influence of other factors

Although the current investigation demonstrates the significant impact of blade surface roughness, turbulence intensity and
time averaging on wind turbine power output, it is imperative to acknowledge that additional variables also play crucial roles. Among these, atmospheric conditions including shear, that has briefly been demonstrated in this paper to significantly influence the performance, as well as temperature, veer, seasonal effects and climate change. Changes in temperature can affect the viscosity of oils and greases, as well as lead to variations in component losses - for instance those in generators and cables - and component stiffness. Other mechanical factors such as, component wear, yaw misalignment, pitch system reliability,
ageing, operations and maintenance events and increased friction in the drive train, significantly influence turbine performance. Moreover, reliable measures of wind speed, necessitating regular calibration of the turbine's wind speed sensor based on turbine output or turbine control programmable logic controller (PLC) parameter or software updates, along with the effects of wind speed binning, are pivotal in evaluating turbine performance accurately. Furthermore, the control of the wind turbine such as generator speed and pitch as a function of wind speed or power, potentially influences the outcomes of such analyses.
Although, these aspects were outside the purview of the present study, they warrant further exploration for a comprehensive understanding of their individual and combined impacts on turbine power output. Future research, building on work such as Malik and Bak (2024), should prioritise a holistic approach to systematically investigate the complex interplay between these factors and their implications for the long-term efficiency and sustainability of wind turbines.

# 4 Conclusion

This study has examined the power and energy losses of multi-megawatt wind turbines caused by erosion-induced degradation of blade leading edges, emphasising the critical role of aerodynamic performance in wind energy capture. A significant aspect of this work has been the use of time-dependent aeroelastic computations to investigate the feasibility of observing the power degradation in real-world measurements. To achieve this, not only were the aerodynamic characteristics degraded, but the influence of turbulence intensity and the time-period for averaging data were also investigated due to their suspected influence on the analysis.

The investigation reveals that blade roughness significantly affects wind turbine performance; yet, it also demonstrates that turbulence intensity significantly masks this degradation. Based on 10-minute averaged data the impact of turbulence intensity on the power is significant, especially in the partial load region, whereas the impact of blade erosion in this region is less pronounced. Notably, blade roughness can significantly affect power production, particularly at wind speeds between 9 and 13 m/s, i.e. in the transition between the partial load region and rated power.

The power coefficient study emphasises the criticality of considering both blade roughness and turbulence intensity when assessing wind turbine performance. It appears that turbulence intensities greater than approximately 10% make the analysis very challenging. The determination of power coefficients and the observation of values exceeding the Betz limit illustrates this.

Findings of the AEP analysis reveal that, for a given site, even mild simulated erosion reduces AEP by 0.82% at 6% $TI$, while more severe erosion leads to a 1.46% decrease. Additionally, the study indicates the variable impacts of erosion and turbulence intensity across different wind climates. In climates characterised by lower average wind speeds, the effects of erosion and turbulence intensity on AEP are accentuated compared to those in wind climates with a higher average wind speed. A key finding from this analysis is that turbulence intensities exceeding 10% may introduce significant uncertainties in power performance analysis. Therefore, when feasible, it is recommended to filter out such high turbulence intensity data to ensure more reliable assessment of wind turbine performance.

Furthermore, the exploration of the influence of time averaging on power output through simulations across different turbulence intensities and time periods provides additional insights. The findings indicate that larger time averaging periods generally result in greater percentage decreases in power, where rising turbulence intensity causes a decrease in power of up approximately 10% for 300 and 600 second periods at 15% $TI$ and 7 m/s wind speed. At the 'knee' of the power curve, at 11 m/s, smaller time periods of 30 and 60 seconds elevate the power curve, while shorter time periods of 1 and 120 seconds have a more neutral effect. Longer time periods of 300 and 600 seconds lower the power curve by up to -4.5% for the latter period, at 15% $TI$ - although it should be noted that higher turbulence intensities are less likely at increased wind speeds. Thus, at 11 m/s, different time periods have both increasing and decreasing effects on power output. This analysis has shown that 10-minute (600 second) time average periods result in values significantly different from those based on smaller time averaging periods. Notably, analysis based on 1-second time periods appear to be neutral to turbulence intensities. Thus, this study indicates that using short time periods results in less influence from turbulence intensity when analysing measurement data.

This study improves the identification of degradation in operational wind turbine measurement data, although many uncertainties remain. Future research should broaden the scope to investigate how leading edge roughness, turbulence intensity, wind shear, seasonal effects, yaw misalignment and other factors such as operations and maintenance events collectively influence annual energy production. This research should focus on the long-term implications of these combined effects and could inform the development of optimised maintenance and operational performance monitoring strategies.

*Code availability.*  The software code used in this study is not publicly accessible due to proprietary restrictions and confidentiality agreements. Despite this, the methodology and algorithms are described in detail within the paper, allowing readers to replicate the results. For third-party code utilised in this study, we have provided references.

*Data availability.*  The datasets analysed in this research are not publicly accessible due to proprietary restrictions and confidentiality agreements. Nonetheless, we have provided a comprehensive description of the data methods within the paper to enable replication by interested researchers.

*Author contributions.*  THM was the primary researcher, responsible for the conception of the study, all experimental work, data collection, analysis and writing the paper. CB, as the PhD supervisor, provided oversight, theoretical support and guidance in refining the research methodology and paper.

*Competing interests.*  The author Tahir H. Malik's PhD was funded by Vattenfall, where he is also employed.

*Acknowledgements.*  We gratefully acknowledge the support of Vattenfall, particularly for financing this study and providing access to vital wind turbine resources. We acknowledge the use of AI language model, OpenAI (2024), for refining a previous version of the manuscript.

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
