# Peer review of "Challenges in detecting wind turbine power loss: the effects of blade erosion, turbulence and time averaging"

_Wind Energy Science, 2024_

## Author Comment (AC1)

**RC1: 'Comment on wes-2024-35', Anonymous Referee #1, 23 Apr 2024**

1. Please take care of some typos and incorrect sentences:

> Corrected

2. In the introduction and conclusions, the authors state the novelty of verifying the effect of TI and erosion on the prediction of AEP and performance losses using an aero-servo-elastic simulation tool. This is not new; similar results and comparisons have been presented, for example, in the following literature:

> This comment is valid in that indeed turbulence as been investigated over time. The current paper, however, does not investigate aerofoil performance as a function of turbulence, but instead focuses on rotor performance. Our primary objective is to understand how mean power output is increased at low wind speeds and decreased around the shoulder of the power curve as a function of various turbulence intensities, in a systematic manner. This aspect of the study is new and we shall make this clear in the revised paper. While turbulence is indeed investigated in the referenced papers, some of them focus on turbulence at the aerofoil level, which is not the focus of our research. However, we acknowledge that Cappugi et al. (2021) partly investigate power curves as a function of turbulence intensity and we shall reference their work appropriately in our paper.
> Furthermore, an answer to this comment can be found in the introduction that has been rephrased.

3. I understand the issue of working with proprietary data and models; however, the authors should indicate more clearly how the presented study and results, which are not replicable as the data from the case study are not disclosed and not generalizable as the results apply to a particular erosion type and distribution and to a specific WT model, can be beneficial for industry or research in this field.

> We acknowledge the reviewer's concern about the proprietary nature of the data and models employed in our study. While ideally, we would have preferred to publish all details, there are significant knowledge bases in data from operational wind turbines that cannot be described in detail due to proprietary constraints. Nevertheless, this should not preclude us from analysing the data and deriving valuable insights.
>
> The trends and relationships identified in our study are highly relevant for the industry. Specifically, wind farm owners and operators, rather than OEMs, face daily challenges reflected in this study, such as lack of access to detailed proprietary information such as aerofoil shapes – a challenge also encountered by the present authors. Understanding these challenges is crucial for researchers to effectively support the industry. Our paper focuses on relative changes in performance rather than absolute values, which is crucial as wind turbine operators should respond to percentage performance reductions rather than reductions measured in megawatts.
>
> Although the specific results pertain to a particular turbine model and erosion type, the approaches and methodologies employed are generalizable. Consequently, the methods and conclusions presented in the paper should be applicable to various types

of wind turbines, advancing the understanding of wind turbine performance under varying conditions.

We shall include a comment in section 2.1 to highlight how this study can contribute to the general understanding of power performance degradation. This addition shall clarify the broader implications and utility of our research for both industry and academic audiences.

4. At pg3: "Although the tested airfoil is not an identical match to that in the HAWC2 model, this approach is deemed a suitable approximation for representing the outboard region of eroded turbine blades." This sentence is not supported by data in the paper; it could be helpful to show that the power curve obtained with the original airfoil matches the power curve obtained with the substituted clean airfoil.

> We appreciate the reviewer's comment regarding the need for clarity on the aerofoil substitution. We did not replace the original aerofoil in the multibody model. Instead, we used wind tunnel data from a different eroded aerofoil to understand the impact on lift coefficient (CL) and drag coefficient (CD). These impacts were then applied as adjustment factors to the original aerofoil characteristics of the proprietary multibody model, specifically for the outer 15% of the blade. This approach allowed us to simulate the effects of leading-edge erosion on turbine performance without altering the fundamental aerofoil design of the model. Therefore, in its clean form, the substituted aerofoil would match the original aerofoil exactly, as would its power curve. Furthermore, we shall include a comment in section 2.2 to clarify this.

5. pg.3 "To represent the effects seen in the wind tunnel experiments, derived factors were used to approximate the results. For simplicity, the lift polar representing the clean airfoil was scaled by a factor of 0.9. Additionally, two artificial drag polars were created by scaling the drag polar representing the clean airfoil by factors of 1.5 and 2.0, respectively." This aspect of the methodology is not very clear and requires clearer explanation in the text. Furthermore, a proper justification should be reported in terms of the scientific soundness of the approach, by reference to existing literature proving the reliability of the approach, or by some verification test or evidence.

> We appreciate the reviewer's feedback regarding the clarity and scientific justification of our methodology. Unfortunately, due to proprietary constraints, we are unable to present the specific polars used in our simulations. To address this limitation, we aimed to represent roughness and erosion effects in a simplified manner.

The goal is to simulate a realistic scenario where erosion has a measurable negative impact on the aerofoil's aerodynamic characteristics (lift and drag coefficients). While we acknowledge that not all aerofoils degrade in the same way, these factors are meant to represent a moderate and severe level of degradation, respectively.

Our approach is based on findings from other wind tunnel tests, which showed that one instance of degradation resulted in a lift reduction to 0.9 times the clean lift coefficient (CL_clean) and drag increases to 1.5 times (CD_clean) and 2.0 times (CD_clean), respectively. This does not imply that all aerofoils exhibit this degradation pattern, but rather it represents one scenario of moderate degradation and one of severe degradation. These factors correspond to (CL/CD)_clean ratios of 0.9/1.5 = 0.6 and 0.9/2 = 0.45, respectively. This ensures a significant degradation effect in our data.

In other words, the reduction factors are chosen to ensure the simulated data reflects a meaningful deterioration in performance, even if the exact values might not perfectly match every real-world case.

The reduction in CL and the CL/CD ratio naturally leads to a reduction in power, which is why these factors are applied. We shall include a comment in section 2.2 of the paper to clarify this reasoning. Additionally, we shall reference appropriate literature to support the validity of this approach, demonstrating that it is a scientifically sound method for approximating the effects of erosion and roughness on aerofoil performance.

We believe this explanation shall enhance the clarity of our methodology and provide the necessary justification for our approach.

6. Figs 7, 8, 10, 12, 13 show many overlapping curves and are very difficult to read. I suggest reducing the number of entries to a subset of the most significant ones.

➢ We appreciate the reviewer's comment regarding the readability of Figs 7, 8, 10, 12 and 13. We acknowledge that we are challenging the reader. In response to your feedback, we shall tried our best to reduce the number of entries in these figures, so as to remove those that are most dispensable. Additionally, the figures are better represented and described. Where highlighting trends, we have enhanced the graphs using colour variations to achieve this. These adjustment shall help enhance clarity and readability, making the data more accessible and easier to interpret.

7. Please improve the clarity of captions for Tables 1 to 5. (From the captions of Table 1 and 2, it is not possible to understand the difference between the data presented in the two tables).

➢ This is a good comment that we understand when we read them again. They shall be changed to:
o Table 1. Change in AEP as a function of TI. Row 2 shows AEP change relative to "clean" performance at TI=6%; Row 3 shows AEP change relative to "P400" performance at TI=6%; Row 4 shows AEP change relative to "P40" performance at TI=6%.
o Table 2. Change in AEP as a function of TI and roughness level: AEP change relative to "clean" performance at TI=6%
o Table 3. Change in AEP as a function of TI and roughness level: AEP change relative to "clean" performance at TI=6% with V_ave=6m/s
o Table 4. Change in AEP as a function of TI and roughness level: AEP change relative to "clean" performance at TI=6% with V_ave=6m/s
o Table 5. Change in AEP as a function of TI and roughness level: AEP change relative to "clean" performance at TI=6% with V_ave=6m/s

8. I understand it is part of the title and the research, however, the discussion about time averaging in the power curve is somewhat off-topic with respect to the other two. Indeed, TI and erosion are features related to the model of the case study, while the time averaging is a data processing parameter. Furthermore, I needed more than one reading to understand the point from section 3.4. I would suggest reducing this part, possibly moving it to an appendix, recalling the main outcome not in the results but in the presentation of the case study and methodology. This will improve the readability of the paper and also simplify the outcomes and conclusions.

> We appreciate the reviewer's comment regarding the discussion on time averaging in the power curve. We believe the averaging of data is very relevant and its inclusion is crucial for understanding the analysis.
>
> The use of 10-minute average values is a standard approach for investigating data from measurements and since our aim is to enhance the understanding of these measurements, we investigated this through simulations.
>
> Our investigation revealed that the impact of time averaging is almost as significant as the effects of roughness itself. Therefore, we believe it is pertinent to include this discussion. We do however acknowledge that the current presentation may challenge readability and that this aspect was not adequately explained.
>
> To address this, we have rephrased the introduction to clarify the rationale behind the analysis of time averaging. As well as rephrasing of section 3.4, with the inclusion of a new Figure 14. The conclusions section is also rephrased.

9. The conclusion section needs some revision. It sometimes presents repeated information ("This research contributes valuable insights into the multifaceted effects of turbulence intensity, blade roughness, and time averaging on wind turbine performance") or conclusions that seem not really related to the data presented ("it highlights how data analysis techniques can either mask or reveal the subtle effects of erosion and turbulence"). Furthermore, a clearer conclusion should be drafted on the discussion about extracting information on erosion-related performance damage from in-field measurements and SCADA data.

> On reflection and in response to this feedback, we have rephrased the conclusion to eliminate redundancies and ensure that it is clearer.

---

## Author Comment (AC2)

**RC2: 'Comment on wes-2024-35', Anonymous Referee #2, 23 Apr 2024**

- Line 1: typo (TI)can -> (TI) can.
➢ Fixed

- Line 5: Erosion and roughness are closely related and sometimes their meanings are mixed up. In the abstract, please specify clearly why erosion is represented in this study with different roughens values.
➢ The abstract has been updated as has the opening sentence of Section 2.2

- Line 20: energy losses of 7% (if it is AEP, please specify it).
➢ Fixed

- Line 30: strategiesBadihià strategies Badihi
➢ Removed sentence due to updated introduction

- Line 30: (2022)Gonzalez -> (2022) Gonzalez
➢ Fixed

- Line 30: The sentence " the lack of insight limits the potential benefits of quantifying APE loss" is difficult to understand, please rewrite it.
➢ Removed sentence due to updated introduction

- Line 35: " protection (LEP) and aerofoils more robust towards the effects of erosion" please give previous examples and references for LEP and robust airfoils.
➢ The following reference has been added to the paper at a different section: Bak C, Anderson P, Madsen HA, Gaunaa M, Fuglsang P, Bove S, Design and Verification of Airfoils Resistant to Surface Contamination and Turbulence Intensity, Conference: 26th AIAA Applied Aerodynamics Conference, August 2008, DOI: 10.2514/6.2008-7050

- Section 2.2. Please in this section it is necessary to specify the % of blade in which roughness is considered. It is 15% but can be found in subsequent sections.
➢ Wording improved to clarify

- Line 79: it is mentioned that the NACA airfoil used in Krog Kruse experiments it is not the same as the one in the HAWC2 model. Please specify if in the computations presented in the paper the airfoil used in the outer part of the blade is the NACA one or the one in the confidential wind turbine model.
➢ Very good point. It has now been reworded to clarify that it was the original proprietary aerofoil to which the factors have been applied, rather than replacement with an alternative aerofoil.

- Line 87: please improve the explanation of the derived factors. If they are used in the airfoil coming from the HAWC2 model it is not very clear specified.
➢ Section 2.2 improved to clarify this point

- Line 94: please specify what is the 'plate behaviour'
➢ Text updated with explanation of meaning

- In Section 2.2 it is not clear whether the control system will be activated during all the simulations performed in this work. That is, describe if the control system detects that the power production is not achieved due to the affected airfoils will make any actions and mitigations.
  - ➤ Section rewritten to clarify

- Line 169: 'with imposed wind shear conditions' : please explain which are these conditions.
  - ➤ Updated text.

- Figure 8: it is difficult to compare since the lines are in top of each other in 2 groups. Maybe a table or separating in different figures for each TI could help. The question that should be answered in this figure is: for all the TI studied the % of power loss is similar or depends on the TI value?
  - ➤ Improved this graph as well as numerous others. Where highlighting a trend we have enhanced the graphs using colour variations to achieve this.

- Line 213: the sentence ' averaging effect of time averaging' is a bit confusing for the reader. Please rewrite it.
  - ➤ Corrected

- Figure 9 caption: change -> Change
  - ➤ Corrected

- Figure 9: all the cases for P40 are computed for different TI, but the reference is always clean and TI 6%: is this consistent?
  - ➤ Doublechecked for consistency

- Figure 12: In the legend 'Clean 6% TI' appears twice, please specify the difference between them in the text and in the legend.
  - ➤ Corrected

- Line 272: It is suggested that the unexpected behavior detected for certain turbulence conditions that is presented in Table 1 is removed for the study. Once it is clarified it could be presented in future works.
  - ➤ Corrected. Turbulence of 20 and 25% are not representative of offshore conditions and have been removed from all data tables.

- Table 1 caption: ' the same profile' -> it is not clear which profile
  - ➤ Caption improved

- Table 1 and Table 2 captions: please include the velocity in this case to be consistent with Tables 3-4-5
  - ➤ Updated

- Line 398: these factors and these aspects appear several times in the paragraph.
  - ➤ Corrected

- Conclusions section: In line 409 it is said that the impact of blade erosion was less significant and right after, 'Blade roughness can significantly affect power-production' Even thought it is ok, do not write them so close because is a bit confusing.
  - ➤ Good point. Section re-worded.

- Please review the whole conclusions section, it is very schematic.
  - ➤ Entire conclusion reworked

- Competing interest sentence: the word 'by' appears twice and is a typo.
  - ➤ Corrected

---

## Author Comment (AC3)

**RC3: 'Comment on wes-2024-35', Anonymous Referee #3, 04 May 2024**

First and most importantly the scientific archival value of the study is not apparent, and perhaps not strong enough for journal publication. Despite the large quantity of data presented in the study the main take away that I was able to see from the study is the fact that differences in power curve and AEP caused by erosion can be similar or even smaller in magnitude than the differences caused by turbulence intensity. This can make it hard to diagnose blade erosion in the real world, as the decrease in power output may be masked by variations in turbulence intensity. This message is interesting, but the large amount of data that is presented in the paper is redundant for this relatively straightforward message. In addition, authors do not suggest ways to work around this issue but the discussion in the paper is limited to presenting the data and little else. Finally, as the authors suggested, the large effect of turbulence on wind turbine power curve can already be found in the existing scientific literature, making the specific contribution of this work somewhat more unclear, and diminishing the scientific value.

> ➤ We appreciate the reviewer's detailed feedback. The primary aim of our research was to develop methods for detecting wind turbine performance degradation, particularly due to blade erosion, in operational turbines using real-world SCADA data. This is a critical issue in the industry, where significant time and resources are spent annually to diagnose such degradation, which has proven challenging due to the lack of an established correlation between blade erosion and full-scale turbine performance degradation.
>
> Furthermore, the extent to which turbulence intensity and time-averaging practices influence the analysed performance, potentially masking the effects of blade erosion, is not well understood. We acknowledge that this message may not have been effectively communicated in the original manuscript.
>
> We respectfully disagree with the assertion that our study lacks scientific value. This topic necessitates further insights so as to better analyse full scale turbine measurement data. Our research contributes to support the enablement of detection of erosion, a challenge recognized by both industry and academia.
>
> To address the reviewer's concerns, we have rephrased both the introduction and conclusion to more clearly articulate the purpose and significance of our study. We emphasize that understanding these influences is essential for accurately diagnosing aerodynamic degradation in wind turbines, a complex task due to the interplay of various factors.
>
> Furthermore, while we acknowledge that the effect of turbulence on wind turbine power curves is well-documented, our study goes beyond this established knowledge. Rather than focusing on aerofoil performance as a function of turbulence, we investigate rotor performance. While turbulence is indeed investigated in existing literature, some of them focus on turbulence at the aerofoil level, which is not the focus of our research. Specifically, our focus is in the context of how turbulence intensity variations can obscure the detection of blade erosion in operational settings.
>
> In response to the reviewer's feedback, we have especially rephrased the introduction and the conclusion section. We believe these revisions and clarifications throughout shall better highlight the scientific value and practical implications of our research for the wind energy industry.

To add to this, the data appears to be presented without a clear goal in mind. This makes the manuscript, despite it being divided in many subsections, very difficult to read. It is unclear what is the "glue" between the sections and how they contribute to the final take aways. In addition, many graphs are hard to understand (such as Figures 7, 13). Some other Figures are hard to tell apart – for instance it took me quite a while to understand the difference between figures 16-1 and 18-19.

> ➢ This adds to the comment above and we recognize the need to be clearer in our message. To address this, we have rephrased sections including the introduction and conclusion.

Figure 11 and especially 10 are misleading, and if I have understood how they are computed, incorrect in my opinion. In fact, the Cp seems to be computed by dividing mean power by mean wind speed. This is incorrect, as Cp is an instantaneous value and should be computed based on instantaneous power and wind speed, and then averaged. Please explain how these values are computed. Authors attempt to warn readers about the high values of Cp in figure 10 at lines 237-241 but the explanation could be improved. The main reason for the large Cp values is the fact that wind turbine power near cut-in as a function of wind speed is cubic, thus increases in wind speed increase power more than decreases in wind speed do.

> ➢ We appreciate the reviewer's comments and concerns. We are fully aware of the reasoning behind the high Cp values and we shall make this issue clearer. We acknowledge the potential for misunderstanding and shall take steps to clarify this issue. The primary message we intend to convey here is that power curves, could be understood as steady-state performance, but the unsteadiness is changing this and that is why the figures are presented to make it clear that power efficiency apparently can be very good but that the reason - in this case - is the turbulence level and that one can make false conclusions.
>
> To address this, we shall rephrase section 3.2.3 to make this clear for the reader.

---

## Referee Report (RR1)

**Challenges in Detecting Wind Turbine Power Loss: The Effects of Blade Erosion, Turbulence and Time Averaging**

Referee report to Manuscript Version 4

- **Line 208:** It seems that a new typo appears 'as it an atmospheric...'

- **General comment 1**: For some figures (example Figure 5 & Figure 8) the blade condition is named as P40 roughness, and for some others (example Figure 11 & Figure 12) the blade condition is named as P40. In some parts of the text (line 220) the blade condition is named as erosion and in some other (example line 299) the blade condition is named as roughness 'the power reduction due to roughness'. This double way of calling the degradation status of the blades could confuse the reader. It is suggested to harmonize the figures and the text an select a way of naming the blade status: for instance blade degradation.

- **General comment 2:** It is explained in section 2.2 that roughness represents a precursor to more significant aerofoil degradation, which is fine but in line 104 it is mentioned that sandpaper provides a simplified model of erosion. The suggestion is to remove the part of line 104 where it is said 'While the sandpaper provides a simplified model of erosion...' and just mention that P40 and P400 degradations are used as reference to calculate the airfoil polar degradation to be used in the aeroelastic model to assess the effect of the different parameters on airfoil performance.

---

## Author Response (AR2)

Dear Peer Reviewer,

Thank you for your valuable and insightful comments on our manuscript. Your feedback has been instrumental in improving the quality and clarity of the paper.

Attached, you shall find a detailed response to each of your comments, along with an updated version of the manuscript reflecting these improvements.

Thank you once again for your thorough review and constructive suggestions.

Best regards,

Tahir Malik

**Report #1: 'Comment on wes-2024-35', Submitted on 04 Jul 2024 - Anonymous referee #1**

I thank the authors for their effort in addressing the required changes, and for their explanation about those suggested changes that they decided to do not apply. The rebuttal is satisfactory, but some of the new text included in the revised manuscript needs attention.

Introduction: The text has been significantly altered and revised. However, the new text sometimes shows poor English, unclear information, and confusing sentences. It still does not clearly explain the novelty and impact of the presented work and results compared to the state of the art.

> ➢ Thank you for your comments, you are right!
> ➢ The novelty of this work has been further emphasised especially in the introduction of the paper.
> ➢ Language sharpened and clarified to reflect the same throughout.

Lines 1-5: The text is complicated. Please simplify the sentence and clarify its meaning.

> ➢ Abstract improved

Lines 40-43: What do the authors mean by measuring AEP losses in a wind tunnel? Please revise this part.

> ➢ Text corrected

Lines 50-52: "Therefore, this study aims to address the following question: 'What makes power losses due to erosion so challenging to detect in operational wind turbines and how can these challenges be more effectively addressed?' The current work addresses this by incorporating a certified model of an operational turbine's controller in the full aero-servo-elastic simulation loop." This text is written with very poor English.

> ➢ You are correct, in response the language has been clarified

Line 55: "The obscurity of erosion's effects" – I am not sure this is the right term to use.

> ➢ Sentence revised

Lines 77-82: Please revise the text and write it in better English (e.g., "it is considered key that a model of a real wind turbine is investigated" is not properly formed).

> ➢ Sentences revised

Figure 1, 2, 5: Revise the captions (e.g., I do not think P40 can be called a "blade condition").

> ➢ Caption text replaced with "blade roughnesses"

Lines 134-135: My understanding is that the data for the clean airfoil of the real WT is available but not disclosable, so the Cl and Cd curves for the original airfoil are available and used to model the nominal WT in HAWK2. The data for the airfoil with roughness are obtained by applying the delta Cl and delta Cd measured on a NACA63 to the polar of the original airfoil. If

this method was applied, I have no further questions on the method, but I would like the authors to revise the text as it is not clearly explained.

- ➢ This is indeed the applied method where multiplication factors are applied to the original aerofoil.
- ➢ The has been improved in response to your comment. The original text could indeed have misinterpreted. Thank you for pointing that out.

Line 195: What are the "factors"?

- ➢ Correct to "shear"

Line 457: Please be more specific about what the "values" are.
- ➢ Specific values included in text. Thank you.

**RC2: 'Comment on wes-2024-35', Submitted on 24 Jul 2024 - Anonymous referee #3**

I find the article has improved with respect to the first version. Particularly the motivation is much clearer in the introduction. To better clarify my previous comment, I was challenging the archival value of this study not because the topic is not a significant issue in the industry, rather because the paper did not provide clear suggestions or best practices beyond presenting the issue. This has now been somewhat been addressed in the conclusions, which present some suggestions and future actions on the topic. I still have some comments:

1) Please revise English carefully throughout. For instance, the first sentence of the revised abstract could be improved "[...] and obscures the underlying reasons." – the subject is missing.

> ➢ Thank you are very right! Abstract improved and other corrections made.

2) Authors are certainly well aware, but perhaps it would not hurt to stress that the influence of turbulence and averaging intervals has been studied previously in the literature even at a rotor level and not only at an airfoil level (for example see https://doi.org/10.1016/j.renene.2020.04.123 (**Saint-Drenan et al. (2020)**) for a summary of some sources). In addition, varying turbulence levels are often cited as a significant source of uncertainty in power curve estimation (IEC 61400-12 and similar). This study does however expand on the topic introducing averaging on different time intervals.

> ➢ This is indeed valuable feedback that is sincerely appreciated. And the work in the field is acknowledged. The following are referenced as important contributions in our paper, particularly emphasizing the role of turbulence on turbine performance.
>
>   o *"This challenge stems from the complex interplay of factors affecting the turbine's performance (Barthelmie and Jensen (2010)), making it difficult to isolate the effects of …"*
>   o As have the following, in the introduction: *"Furthermore, turbulence is a well-known atmospheric condition that significantly impacts wind turbine performance (St. Martin et al. (2016); **Saint-Drenan et al. (2020);** Kim et al. (2021); Cappugi et al. (2021))*
>   o *This alignment with preceding studies \cite{wagner2010simulation} and \cite{saint2020parametric} further validates the critical nature of TI in such analyses.*
>
> ➢ Additionally, the novelty of this work has been further emphasised especially in the introduction of the paper and in the language.
> ➢ In addition to time averaging the study importantly expands **erosions** relative impact, an aspect that has not been extensively addressed in previous work.

3) I understand why the Cp graphs are presented the way they are, but I still find it a bit misleading. I think this discussion could be avoided completely if the equation through which the Cp is computed is reported, clearly showing that averaging is performed on wind speed and power separately, before Cp is computed based on average values.

> ➢ You are correct. The text has been updated for clarity on method of computing Cp, including the used equation and explanation.

4) Section 3.4 is very hard to read, I would strongly suggest revising it completely. It contrasts quite a lot with the introduction, which flows much better and is much easier to understand. The influence of averaging interval on performance decrease comes across quite effectively. It is unclear to me why the 1 second averaging interval seems to greatly reduce the variability in mean power output caused by TI for the clean blade but not in the eroded blades. Elaborating more clearly on this is crucial for the conclusions (particularly in their revised form), as, if I'm not mistaken, authors suggest to use one-second averaging interval

- ➤ Your comment is fair. In response:
- ➤ The sections are better, structured, organised and titled and the baseline for each Figures case has been more clearly articulated.
- ➤ **Importantly,** I had overlooked the 0.01 second time interval for Figures 15, 16, 17, 18, 19 and 20. These have now been added to the figures – **apologies** for the oversight.
- ➤ These figures have been improved using colormaps to show the trend and differentiate between sets of figures and dimensions. Roughly (not strictly) speaking e.g. Copper for turbulence, Winter (blue to green) for time interval, Abyss (blue variation) for roughness
- ➤ **The text has been majorly revised for clarity.**

It is hope that correction of the omissions and improvements in text add to the clarity that caused misunderstanding and correctly support the conclusions. Thank you again for catching this!

---

## Author Response (AR3)

Dear Peer Reviewer,

Thank you for your valuable and insightful comments on our manuscript. Your feedback has been instrumental in improving the quality and clarity of the paper.

Attached, you shall find a detailed response to each of your comments, along with an updated version of the manuscript reflecting these improvements.

Thank you once again for your thorough review and constructive suggestions.

Best regards,

Tahir Malik

**Report #1: 'Comment on wes-2024-35', Submitted on 29 Oct 2024 - Anonymous referee #3**

Dear Authors,

I find the clarity of the manuscript has greatly improved and you have addressed all of my comments.

> ➢ Thank you for your comment!

Unfortunately I believe the quality of writing still needs to be improved. Some paragraphs in the conclusions have especially stood out to me.

> ➢ With tried eyes, you correctly point out that text was in places left needing correction and improvement. I have gone through the entire text and weeded out the text that I had previously overlooked. I hope I have not missed anything!

For example the paragraph starting at line 464 does not flow correctly and the use of present continuous in the term "investigating" is incorrect.

> ➢ Corrected

Also: L462: maybe you mean "criticality"? but I would strongly suggest rephrasing.

> ➢ Corrected

The following phrases up to L475 could also be improved, as past tense is used while present is often used in other parts of the conclusions.

> ➢ **You are correct!** I have implemented improvements in the tense.

Moreover, L489: increasing -> increased

> ➢ Corrected

And, L490: plase choose betweeen "time average" and "time averaging" throughout the paper. "time averagin period" is more appropriate in my opinion.

> • Thank you for highlighting the need for consistent terminology regarding "time average," "time averaging," and "time averaging period." Upon review, I agree that "time averaging **period**" is more precise when referring to the temporal duration of the averaging process, as seen in the example:: "Shorter time averaging periods may allow for detecting the nuanced effects of turbulence and erosion more effectively."

To ensure clarity and consistency throughout the paper the following is the guiding principle:
1. **"Time averaging period"** refers to the **duration** over which data is smoothed or aggegated.
2. **"Time averaging"** describes the **process** of applying the smoothing or averaging technique.
3. **"Time average"** represents the **resulting statistical value** derived from the process.

Additionally, thanks to your highlighting of this terminology: the following clarification of the usage of the terms:

- **Period** implies a specific, well-defined length of time over which the averaging occurs e.g. "The data were averaged **over a period of 10 seconds**."
- **Interval** can also be used, but it often suggests a range between two points in time rather than the length of the averaging period itself, such as in the example: "The **sampling interval** was 1 minute, but the **averaging period was 1 hour**."

**Thank you for your valuable input! The Figures are also updated to reflect this.**

Please revise English throughout the entire manuscript carefully.

➢ Thank you for your comment. Typos and slips were indeed found in phrasing throughout the paper. **I hope I have sufficiently squashed these!**

**Report #2: 'Comment on wes-2024-35', Submitted on 18 Nov 2024 - Anonymous referee #2**

The work makes an analysis of the effect over power production of erosion and turbulence intensity also including the impact of time averaging in the results. The work gives more insight into the complexity of power losses estimation, especially when turbulence intensity appears. The idea of isolating the effect of the different causes is well addressed. The authors attended the first reviews and provided a final version of the manuscript. After reviewing the manuscript, a typo and two general comments are detected and forwarded to the authors.
Referee report to Manuscript Version 4

1) **Line 208:** It seems that a new typo appears 'as it an atmospheric...'.

- ➤ Corrected!
- ➤ Furthermore, as correctly pointed out, also by the other reviewer, typos and slips were found in phrasing. **I hope I have sufficiently squashed these!**

2) **General comment 1:** For some figures (example Figure 5 & Figure 8) the blade condition is named as P40 roughness, and for some others (example Figure 11 & Figure 12) the blade condition is named as P40. In some parts of the text (line 220) the blade condition is named as erosion and in some other (example line 299) the blade condition is named as roughness 'the power reduction due to roughness'. This double way of calling the degradation status of the blades could confuse the reader. It is suggested to harmonize the figures and the text an select a way of naming the blade status: for instance blade degradation.

- ➤ Thank you for the comment – this was inconsistent. Neither was it aesthetically pleasing and additionally could lead to reader confusion, **therefore Figures 5, 8, 11, 12 etc. have all been aligned.** The term roughness has been removed for harmonisation and streamlining of the figures. **Thank you!**
- ➤ Furthermore, although P400 and P40 grid sandpaper roughness is used as a proxy for blade erosion. Leading edge erosion (LEE) and leading edge roughness (LER) are used interchangeably as is the case in the field.
- ➤ Despite this, your comment is very fair -I have gone over the entire paper **to ensure that it is not used in *too* interchangeable a manner for consistency**. Yet, some instances remain due to its proxy usage for erosion. Examples are:
  - o Abstract:" aerofoil characteristics for the blade were modified to simulate different degrees of erosion, represented by varying levels of roughness."
  - o "Blade leading edge erosion was modelled as varying levels of surface roughness, a quantifiable measure of damage severity"
  - o Erosion is the degradation represented by P400 and P40 roughness

3) **General comment 2:** It is explained in section 2.2 that roughness represents a precursor to more significant aerofoil degradation, which is fine but in line 104 it is mentioned that sandpaper provides a simplified model of erosion. The suggestion is to remove the part of line 104 where it is said 'While the sandpaper provides a simplified model of erosion...' and just mention that P40 and P400 degradations are used as reference to calculate the airfoil polar degradation to be used in the aeroelastic model to assess the effect of the different parameters on airfoil performance.

- ➤ Thank you. The paragraph has been amended to reflect your fair comment.